**Data Availability Statement:** All relevant data are within the manuscript and its Supporting Information files.

**Funding:** The author(s) received no specific funding for this work.

# Knowledge, attitude, and practice on antibiotic use and antibiotic resistance among the veterinarians and para-veterinarians in Bhutan

**Karma Wangmo**[1]*, **Thinley Dorji**[2], **Narayan Pokhrel**[3], **Tshering Dorji**[4], **Jambay Dorji**[4], **Tenzin Tenzin**[5]

1 Department of Livestock, Regional Livestock Development Centre, Trashigang, Bhutan, 2 Kanglung Hospital, Ministry of Health, Trashigang, Bhutan, 3 Department of Livestock, District Veterinary Hospital, Pemagatshel, Bhutan, 4 Department of Livestock, Regional Livestock Development Centre, Zhemgang, Bhutan, 5 Department of Livestock, National Centre for Animal Health, Thimphu, Bhutan

* karmazwank@yahoo.com

## Abstract

### Introduction

Antimicrobial resistance is (AMR) an emerging global public health problem. Rationale use of antibiotic can prevent the rise of antimicrobial resistance. The objective of this study was to understand the knowledge, attitude and practice on antibiotic usage and AMR among the veterinarians and para-veterinarians in Bhutan.

### Method

A cross-sectional questionnaire survey among the veterinarians and para-veterinarians was conducted from June to July 2020. A score of one to the correct answers and zero for the wrong answers was allotted to each respondent answers. The total score was added and those who scored above the mean was categorized as having good knowledge and favourable attitude.

### Result

A total of 219 animal health workers participated in this study. The mean knowledge score was 12.05 ±1.74 with 38.8% of the respondents having good knowledge on antibiotic use and AMR. Similarly, the mean scores for the attitude level were 8.32±1.61 with 51% them having favorable attitude towards antibiotic usage and AMR. The mean practice score was 3.83±1.06 with 77% of them having good practices on antibiotic use. The respondents who read national plan on AMR were found to have good knowledge on antibiotics and AMR (AOR: 2.39; 95% CI: 1.19–4.82). The female respondents (AOR: 2.16; 95% CI: 1.01–4.61), respondents from the eastern region (AOR: 2.53; 95% CI: 1.18–5.44), west central (AOR: 3; 95% CI: 1.30–6.92), animal health supervisors (AOR: 9.77; 95% CI: 1.98–48.29), and livestock production supervisors (AOR: 2.77; 95% CI: 1.21–6.35) have favorable attitude towards antibiotics and AMR.

**Competing interests:** The authors have declared that no competing interests exist.

## Conclusions

Our study identified that most animal health workers in Bhutan had poor knowledge on antibiotics usage and AMR. Therefore, regular awareness education on antibiotics and AMR in the form of refresher course/training must be provided to the animal health workers in the country to avoid inappropriate use of antibiotics.

## Introduction

Antimicrobial resistance (AMR) is a multifaceted global public health issue affecting both human and animal populations [1]. The over and under prescription, and misuse of antibiotics lead to development of antimicrobial resistance [1]. All bacteria have the potential to develop resistance against all available antibiotics, which will eventually lead to non-availability of effective antibiotics to treat even common infections. It is estimated that by 2050 approximately 10 million people will die annually due to infection related to antibiotic resistance [2]. Inappropriate use of antibiotics in both humans and animals and their use as growth promoter in animals has led to the development of resistance [3]. There is adequate evidence that show positive correlation between inappropriate antibiotic usage and the speed of resistant development [4, 5]. Therefore, prudent use of antibiotics in both animal and human health system is essential to prevent resistance development [4].

Antimicrobial stewardship programs at different levels of the system nationally and internationally plays crucial role in combating antimicrobial resistance. To address the emergence of AMR, World Health Organization (WHO) initiated a global action plan on AMR (GAP-AMR); one of which is to enhance knowledge and surveillance skills on AMR through global antimicrobial resistance surveillance system (GLASS). Bhutan is one of the member country of GLASS and has been actively involved in enhancing laboratory capacity to perform antibiotic susceptibility testing (AST) and AMR surveillance system in the country [6].

In Bhutan, the increased public demand for animal products and the government's policy of self-sufficiency has led to increase in livestock enterprises especially poultry, piggery and dairy farming [7, 8]. This has led to increased use of antibiotics for therapeutic, prophylactic and metaphylactic treatments. In Bhutan, all animal health services including medicines and drugs are provided free of cost by the government. The farmers are highly dependent on government animal health workers for all kinds of animal health services and medicines. Therefore, the animal health workers play a critical role in use of antibiotics in animals [9]. Nevertheless, the sale and distribution of antimicrobials both in human and animals are regulated under the medicine rules and regulation and are enforced by Drug Regulatory Authority (DRA) of Bhutan [10]. The commonly used antibiotics in animal health in Bhutan are aminoglycosides, cephalosporins, macrolides, penicillins, quinolones sulphonamides, tetracyclines and metronidazole [9]. A few studies in Bhutan have shown the existence of pathogens resistant to various antibiotics and some of the pathogens were of zoonotic importance [11].

Although, there are no published study regarding knowledge, attitude and practice (KAP) on antibiotic usage and AMR among animal health workers in Bhutan, studies from other countries indicate the importance of KAP of animal health workers to understand the level of their knowledge in judicious use of antibiotics and prevention of resistant development [5]. For instance, a study among Dutch veterinarians revealed that veterinarians with positive attitude and favourable knowledge on AMR had positive impact on antimicrobial use (AMU) and positive influence among farmers [12]. Similarly, a study in Nigeria showed that there was little awareness on AMR among veterinarians and the role and use of biosecurity and prophylactic

antibiotic use in the prevention of infection was poorly understood [13]. A study on KAP associated with AMR and AMU among veterinary students in Nigerian University and among university students on antibiotic usage in Nepalese University had revealed unsatisfactory knowledge on AMR and AMU among students. These studies have suggested intervention measures such as workshops, seminar and campaigns on antibiotic use and AMR in the universities to increase knowledge and awareness level among the students [14, 15].

Therefore, the main objective of this study was to assess the knowledge, attitude and practice (KAP) associated with antimicrobial usage and AMR among animal health workers working under different government organization in Bhutan. The findings from this study will be useful to get information on antibiotic prescribing trends among veterinarians and para-veterinarians, which will help in designing an evidence-based education and advocacy plan on prevention of antimicrobial resistance in the country.

## Materials and methods

### Study area

Administratively, Bhutan is divided into 20 districts (dzongkhags) and 205 sub-districts (gewogs). Each district has one district veterinary hospital (DVH), and each sub-district has one livestock extension centres (LEC) which provide animal health services to the public/communities. In addition, four city veterinary hospitals (CVH) and one National Veterinary Hospital (NVH) provides animal health services in the urban areas. The DVH, CVH and NVH are manned by veterinarians while LECs at the sub-districts are manned by para-veterinarians. The veterinarians at the district veterinary hospitals provides technical supervision of para-veterinarians working in the sub-district level. In this study, we included all government veterinarians and para-veterinarians working at the district veterinary hospitals, city veterinary hospitals and livestock extension centres that provide animal health services in Bhutan. The para-veterinarians in Bhutan have a formal two years diploma in animal husbandry (after completion of grade 10 and 12) where they are taught veterinary medicines and pharmacology in their curriculum. They are also provided refresher course in veterinary medicine and veterinary drugs time to time. They also seek technical support from the veterinarians and refer the Drug Formulary developed by the Department wherever necessary. In addition, the Drug Regulatory Authority of Bhutan conduct constant monitoring of the antibiotic prescription in the field.

### Questionnaire design and data collection

This was a cross–sectional study and the data was collected using a self-administered structured questionnaire. The questionnaire comprised of five sections: the first section consisted of the demographic information of the respondents, second section contained 17 questions on knowledge on antibiotics and AMR, source of knowledge and four questions on awareness on AMR, third section had 11 questions on attitudes and fourth section had five questions on practice with three case scenarios on antibiotic use. The fifth section included questions on prescribing pattern of antibiotics. The questionnaire was adapted from similar studies conducted elsewhere on KAP of antibiotics and AMR [16–18]. The questionnaire was initially pre-tested among 15 para-veterinarians and two veterinarians to check for the validity and were modified accordingly. These group of people were later excluded from the main survey. The questionnaire was self-administered online using google forms from 1st June to 31st July 2020. The online link was shared with the government veterinarians and para-vets working in the field, and were handling/prescribing antibiotics for the treatment of animal cases. Respondents could participate in the survey voluntarily and had the opportunity to withdraw from

participation at any time. There were only 312 eligible animal health workers in the country, so all were invited for the survey.

## Data analysis

The data was downloaded from the google form and checked for any missing answers. Data analysis was performed using STATA version 13 (Stata Corporation, College Station, TX, USA) software. Descriptive statistics were performed by calculating the proportions and frequencies to describe the demographic characteristics and KAP scores. A score was allotted to each respondent answers to questions on knowledge and practices. For every correct answer to each question, a score of "1" was allotted and '0' for the wrong answers. The total score was added and those who scored above the mean (12.05 ±1.74) was categorized as "good" and those who scored equal to and below mean was defined as "poor" knowledge on antibiotics use and AMR. Similarly, for the attitude related questions, a score of "1" was allotted for the correct answers and "0" for the wrong answers. If the attitude scores were more than or equal to mean (8.32±1.61) it was considered as "favorable attitude" and if the scores were less than mean score, it was considered as "unfavorable" towards antibiotics use and AMR.

First, a univariable logistic regression analysis was performed to assess the association between the binary outcome variables (good vs poor knowledge and favourable vs unfavoutable attitude) of the respondents on antibiotics usage and AMR with the independent variables such as age, gender, level of qualification and the type of practice institutions of the respondents. The variables with a p-value of $\leq 0.25$ were selected for the final multivariable logistic regression model. The variables with p-value $\leq 0.05$ in the final model were considered significant. The goodness-of-fit for the final model was assessed using log likelihood ratio test.

## Ethical approval

The study was approved by the Research and Extension Division of the Department of Livestock, Ministry of Agriculture and Forest, Bhutan on 21 April 2020 via letter number DoL/RED-142/2019-2020/1036. Those who responded to the online questionnaire were considered to have provided an informed consent. The response to the questionnaire were stored in the computer with password and were accessible only to the investigators.

## Results

### Demographic characteristics of the respondents

Of the 312 eligible candidates invited for the survey, 221 of them responded. However, two of the respondents logged in for the survey but they did not respond to any of the questions and were excluded from the survey. So, the overall response rate in this study was 70% (219/312).

Of the total respondents, 176 (80%) of them were male and 43 (20%) were female. About 40% of the respondents were in the age group of 20–30 years. Most of the respondents (67%, 148/219) had diploma in animal science and were working in Gewog extension centers (68%, 150/219) as extension supervisors (44%, 97/219). Only 55 (25.1%) and four (1.8%) respondents had degree in either veterinary science (for veterinarians) or animal science (for para-veterinarians) and master's qualifications, respectively. The respondents for this survey were diverse and included from different organization, regions, districts, qualifications, and job positions (Table 1).

**Table 1. Demographic characteristics of the respondents.**

| Characteristic | Frequency | Percentage |
|---|---|---|
| *Gender* | | |
| Male | 176 | 80.4 |
| Female | 43 | 19.6 |
| *Age-groups* | | |
| 20–30 years | 88 | 40.2 |
| 30–40 years | 87 | 39.7 |
| 40–50 years | 27 | 12.3 |
| > 50 years | 17 | 7.8 |
| *Qualifications* | | |
| Certificate | 12 | 5.5 |
| Diploma | 148 | 67.6 |
| Bachelor | 55 | 25.1 |
| Masters | 4 | 1.8 |
| *Organization* | | |
| Dzongkhag Veterinary Hospital | 56 | 25.6 |
| Gewog Extension Center | 150 | 68.5 |
| National Veterinary Hospital | 9 | 4.1 |
| City Veterinary Hospital | 4 | 1.8 |
| *Regions* | | |
| East | 77 | 35.2 |
| West | 59 | 26.9 |
| East Central | 24 | 11 |
| West Central | 59 | 26.9 |
| *Current position* | | |
| Animal Health supervisor | 13 | 5.9 |
| Extension Supervisor | 97 | 44.3 |
| Gewog Livestock Extension Officer | 41 | 18.7 |
| Livestock Production Officer | 4 | 1.8 |
| Livestock Production Supervisor | 43 | 19.6 |
| Veterinarian | 21 | 9.6 |
| *No. of years in service* | | |
| 0–5 years | 69 | 31.5 |
| 6–10 years | 51 | 23.3 |
| 11–20 years | 67 | 30.6 |
| >20 years | 32 | 14.6 |

## Level of knowledge on antibiotics and AMR

Three-fourth of the respondents (74.6%, 162/217) correctly answered that antibiotics are used to treat bacterial infection while 23%, 50/219) of them mentioned that it could be used to treat all types of infections. All the respondents (100%) have answered that antibiotics should be administered with correct dose and dosage for any animal species. Most of the respondents (90%, 208/219) mentioned that the antibiotic course should be continued even if the patient is clinically improved. The majority (95%, 208/219) of the respondents have mentioned that antibiotics should not be used for promoting growth in animals. More than 90% of the respondents mentioned that withdrawal periods should be followed after treating animals with antibiotics for both meat (96.8%, 212/219) and milk (94.1%, 206/219). When asked about the

duration of antibiotics, 78% (169/216) and 49% (107/217) correctly stated that antibiotics should be given for 5–7 days in large animals and poultry (Table 2, S1 Table).

The overall mean knowledge score of the respondents was 12.05 ±1.74 (range 6–17). Of the total respondents, only 39% (85/219) had good knowledge (above mean) while 61% (134/219) had poor knowledge on antibiotic use and AMR. The overall knowledge score was higher for those with bachelors/masters compared to those with certificate/diploma (mean 12.59 vs 11.85; p-value 0.005). There was statistically significant difference in their response between certificate/diploma and bachelors/masters when they were asked if "resistant bacterium can be spread between animals and also to humans". The respondents with bachelors/masters had better knowledge on this question compared to those with certificate/diploma (80% vs 50%).

More than half of the respondents (56%, 122/218) mentioned that any bacteria will become resistant once exposed to an antibiotic while 95% (209/219) thought that animal infected with resistant bacteria will be difficult to treat. More than half of the (58%, 127/219) respondents mentioned that resistant bacterium will spread among animals and to humans. A half (54%, 119/219) of the respondents knew that good management practice will prevent antimicrobial resistance.

## Source of knowledge on antibiotic use and AMR

The major source of knowledge on antibiotic use and AMR in our study was from training programs (38%, 144/219) (Fig 1). This was followed by internet and social medias (19.7%, 102/219). The least mentioned source of knowledge was from radios (5.9%, 13/219)

## Awareness on AMR among the respondents

When asked about whether they have any knowledge on AMR, 98% (215/219) of the respondents had heard of antimicrobial resistance (Fig 2). However, only 14% (31/219) of them had attended refresher course on AMR. Likewise, only 15% (34/219) of them had participated in antibiotic awareness week. More than one-third (37%, 82/219) had read Bhutan national plan on AMR.

## Attitude towards antibiotic use and AMR

The overall mean score for the attitude level of the respondents was 8.32±1.61 (range 3–11). Of the total respondents, 112 (51.1%) had favorable attitude on antibiotic usage and AMR.

The attitude of respondents towards antibiotic use is shown in Fig 3 and S2 Table. When asked about whether a thorough examination of animal was required before the prescription of antibiotics, 96% (211/219) agreed on it. Likewise, three fourth (75%, 165/219) of the respondents agreed that antibiotic sensitivity test is required before administering an antibiotic. The majority (72%, 158/219) of the respondents believed that broad spectrum antibiotics were the right choice for any bacterial infections while 15% (33/219) of them disagreed on this. Most of the respondents (96%, 211/219) agreed that antibiotic resistance is a serious problem worldwide. However, only 59% (131/219) mentioned that antibiotic resistance is a problem in Bhutan. The majority (88%, 193/219) of the respondents strongly agreed on inappropriate use of antibiotics as a cause of resistance. Administration of correct dose and dosage of antibiotics as means of preventing resistance was correctly answered by 91% (201/219) of the respondents. Most of the respondents (88%, 194/219) acknowledge that antibiotic resistance will affect everyone's health. However, there was no significant difference in attitude level among those with bachelors/masters and certificate/diploma (mean: 8.36 vs 8.31; p-value-0.86)

**Table 2. Knowledge on antibiotic use and AMR among the veterinarians and para veterinarians.**

| Characteristic | Frequency (%) | Certificate/diploma (%) | Bachelor/Masters (%) | p-value |
|---|---|---|---|---|
| **Antibiotic are prescribed for** | | | | |
| Virus infection | 1(0.5) | 1 (0.6) | 0 | 0.525 |
| Bacterial infection* | 162(74.7) | 118 (74.7) | 44 (74.6) | |
| Fungal infection | 3(1.4) | 3 (1.9) | 0 | |
| Protozoal infection | 1(0.5) | 0 | 1 (1.7) | |
| All the above | 50(23) | 36 (22.8) | 14 (23.7) | |
| **Antibiotics should be administered with correct dose and dosage for all type of animal species** | | | | |
| Yes* | 219(100) | 160 (100) | 59 (100) | |
| No | 0 | 0 | | |
| **The antibiotic treatment should be stopped once the animal stops showing signs of disease even if the course is not completed** | | | | |
| Yes | 6(2.8) | 4 (2.5) | 2 (3.4) | 0.631 |
| No* | 208(95.4) | 151 (95) | 57 (96.6) | |
| Don't know | 4(1.8) | 4 (2.5) | 0 | |
| **Giving antibiotics to animals that are not sick will prevent it from becoming sick in the future** | | | | |
| Yes | 13(5.9) | 10 (6.3) | 3 (5.1) | 0.917 |
| No* | 200(91.3) | 146 (91.3) | 54 (91.5) | |
| Don't know | 6(2.7) | 4 (2.5) | 2 (3.4) | |
| **If one animal in a herd is sick, all other animals in the same herd should be given antibiotics to prevent infection** | | | | |
| Yes | 20(9.1) | 13 (8.1) | 7 (11.9) | 0.616 |
| No* | 197(90) | 145 (90.6) | 52 (88.1) | |
| Don't know | 2(0.9) | 2 (1.3) | 0 | |
| **Antibiotics should be given to promote growth in animals** | | | | |
| Yes | 6(2.7) | 5 (3.1) | 1 (1.7) | 0.748 |
| No* | 208(95) | 152 (95) | 56 (94.9) | |
| Don't know | 5(2.3) | 3 (1.9) | 2 (3.4) | |
| **Broilers treated with antibiotics should NOT be slaughtered for meat purpose until the completion of withdrawal period of that antibiotic.** | | | | |
| Yes* | 212(96.8) | 154 (96.3) | 58 (98.3) | 0.816 |
| No | 3(1.4) | 3 (1.9) | 0 | |
| Don't know | 4(1.8) | 3 (1.9) | 1 (1.7) | |
| **Milk and milk products from a cow treated with antibiotics can be consumed during the course of treatment** | | | | |
| Yes | 10(4.5) | 8 (5) | 2 (3.4) | 0.663 |
| No* | 206(94.1) | 149 (93.1) | 57 (96.6) | |
| Don't know | 3(1.4) | 3 (1.9) | 0 | |
| **What is the average duration of antibiotic course in large animal?** | | | | |
| 1 day | 1(0.5) | 0 | 1 (1.7) | 0.214 |
| 3 days | 32(14.8) | 24 (15.2) | 8 (13.8) | |
| 5 days* | 132(61.1) | 91 (57.6) | 41 (70.7) | |
| 7 days* | 37(17.1) | 31 (19.6) | 6 (10.3) | |
| 10 days | 9(4.2) | 8 (5.1) | 1 (1.7) | |
| Don't know | 5(2.3) | 4 (2.5) | 1 (1.7) | |
| **What is the average duration of antibiotic course in poultry birds?** | | | | |
| 1 day | 5(2.3) | 5 (3.2) | 0 | |
| 3 days | 72(33.2) | 52 (32.9) | 20 (33.9) | 0.177 |
| 5 days* | 73(33.6) | 53 (33.5) | 20 (33.9) | |
| 7 days* | 34(15.7) | 20 (12.7) | 14 (23.7) | |
| 10 days | 1(0.5) | 1 (0.6) | 0 | |
| Don't know | 32(14.8) | 27 (17.1) | 5 (8.5) | |
| **Any bacteria will become resistant once it is exposed to an antibiotic** | | | | |
| Yes* | 122(56) | 93 (58.5) | 29 (49.2) | 0.353 |
| No | 80(36.7) | 56 (35.2) | 24 (40.7) | |
| Don't know | 16(7.3) | 10 (6.3) | 6 (10.2) | |

*(Continued)*

**Table 2.** (Continued)

| Characteristic | Frequency (%) | Certificate/diploma (%) | Bachelor/Masters (%) | p-value |
|---|---|---|---|---|
| **An animal infected with resistant bacteria will be difficult to treat.** | | | | |
| Yes* | 209(95.4) | 151 (94.4) | 58 (98.3) | 0.32 |
| No | 6(2.7) | 6 (3.8) | 0 | |
| Don't know | 4(1.8) | 3 (1.9) | 1 (1.7) | |
| **A resistant bacterium can be spread between animals and also to humans** | | | | |
| Yes* | 127(58) | 80 (50) | 47 (79.7) | <0.001** |
| No | 48(21.9) | 43 (26.9) | 5 (8.5) | |
| Don't know | 44(20.1) | 37 (23.1) | 7 (11.9) | |
| **Practicing good animal hygiene will prevent development of AMR** | | | | |
| Yes* | 119(54.3) | 88 (55) | 31 (52.5) | 0.83 |
| No | 86(39.3) | 61 (38.1) | 25 (42.4) | |
| Don't know | 14(6.4) | 11 (6.9) | 3 (5.1) | |
| **A cow with a recurrent mastitis is brought to your clinics. The cow was previously treated with penicillin intramammary infusion. What antibiotics will you prescribe now?** | | | | |
| Penicillin intramammary infusion | 29(13.4) | 24 (15.1) | 5 (8.6) | 0.188 |
| Gentamicin injection | 53(24.4) | 36 (22.6) | 17 (29.3) | |
| Cefoperazone intramammary | 49(22.6) | 39 (24.5) | 10 (17.2) | |
| Conduct AST on milk samples* | 77(35.5) | 52 (32.7) | 25 (43.1) | |
| No need for antibiotics | 9(4.2) | 8 (5) | 1 (1.7) | |
| **Three birds in a flock of 500 showed signs of greenish diarrhoea. The birds are eating normal and active. What antibiotics will you prescribe?** | | | | |
| Tetracycline for all the birds | 103(47.5) | 78 (49.1) | 25 (43.1) | 0.123 |
| Tetracycline for the three sick birds | 41(18.9) | 30 (18.9) | 11 (19) | |
| Sulphadiazine for all the birds | 28(12.9) | 23 (14.5) | 5 (8.6) | |
| Sulphadiazine for the three sick birds | 20(9.2) | 15 (9.4) | 5 (8.6) | |
| No need for any antibiotics* | 25(11.5) | 13 (8.2) | 12 (20.7) | |
| **An owner complaint of his bull having fever and inappetence for three days. There is salivation but mucus membranes are pale pink. What antibiotic will you prescribe?** | | | | |
| Oxytetracycline | 75(34.4) | 59 (37.1) | 16 (27.1) | 0.618 |
| Amoxycillin | 16(7.3) | 10 (6.3) | 6 (10.2) | |
| Gentamicin | 6(2.8) | 5 (3.1) | 1 (1.7) | |
| Sulphadimidine and trimethoprim combination | 36(16.5) | 27 (17) | 9 (15.3) | |
| Streptopenicillin. | 13(6) | 9 (5.7) | 4 (6.8) | |
| No need any antibiotics* | 72(33) | 49 (30.8) | 23 (39) | |

**Note:** the answers marked as (*) are the correct answers; others are incorrect/wrong answers. (**) indicates chi square test.

## Practice on antibiotic use and AMR

The mean practice score of the respondents was 3.83±1.06 (range:1–5). Two-third of the respondents (146/219) had good practices on antibiotic use while 33% (73/219) had poor practices on antibiotic use. Although, the overall level of practice did not differ between bachelors/master and certificate/diploma, there was significant difference in duration of antibiotics prescribed. Those holding certificates/diplomas prescribed antibiotics for longer or shorter duration than the optimal duration of 5–7 days compared to the respondents with bachelors/master level qualification.

Prolonged illness with underlying bacterial infections was correctly mentioned by 62% (137/219) of the respondents as indications of antibiotic use (Table 3). However, 24 respondents (11%) mentioned that they used antibiotic for all types of cases. One hundred and eighty

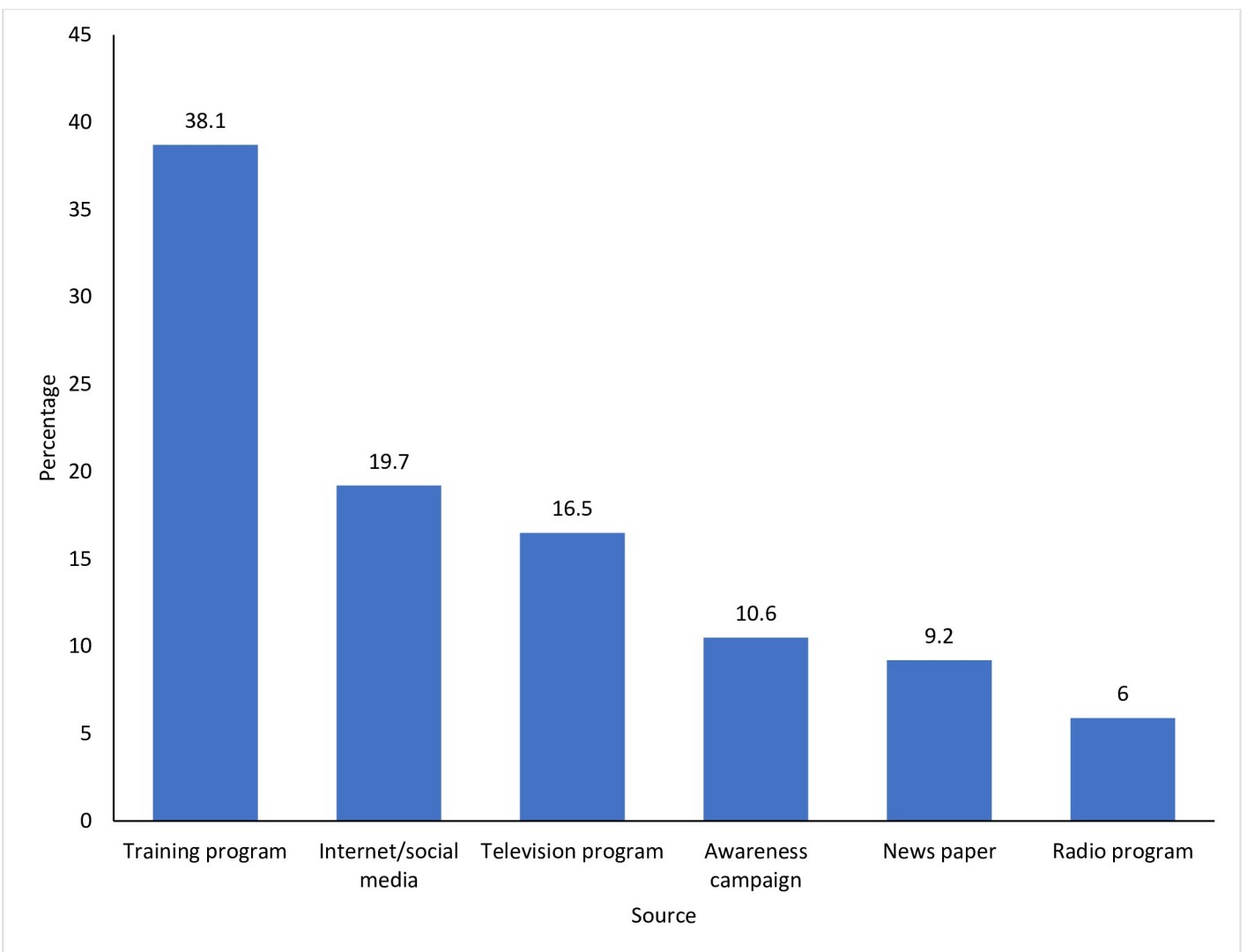

**Fig 1. Source of information on antibiotics and AMR among the 219 respondents (animal health workers in Bhutan).**

respondents (82.6%) mentioned that they made their own judgement for prescription of antibiotic and were not influenced by owner's expectations. While most of the respondents (63%, 138/219) prescribed antibiotics for 5 days, a few of them prescribed for one day (2/217) or till the animal is recovered (4/217). The selection of the antibiotic was done based on the type of bacteria involved by most of the respondents (88.4%, 187/219).

### Prescribing pattern of antibiotics by the veterinarians and para-veterinarians

The most commonly used antibiotic by the respondents were oxytetracycline (22%), followed by tetracycline (16.8%) and benzathine penicillin (15%). The least used antibiotics was sulphonamides (5.7%) (Fig 4).

In terms of prescribing pattern, almost 49% (106/216) have mentioned that they prescribed antibiotic more than once a week while 19% reported that they do on daily basis. However, it is worrying to note that only 14% (32/219) of the respondents were completely confident in prescribing antibiotic while 45% (100/215) were fairly confident. Of this, 148 (68%) mentioned

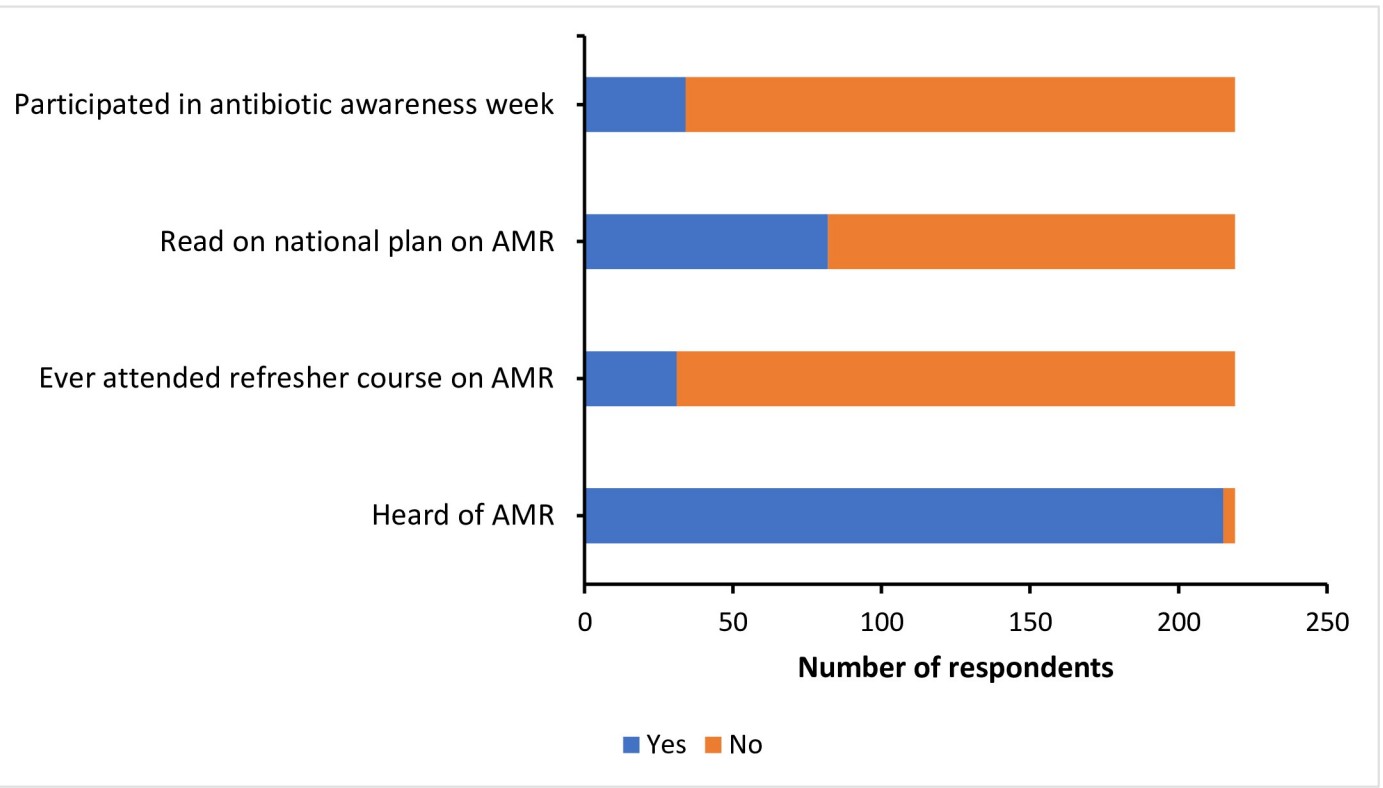

**Fig 2. Level of knowledge and awareness on antibiotics and AMR among the 219 respondents (animal health workers in Bhutan).**

that they were completely or somewhat confident in calculating the dosage of antibiotic for a particular species of animals. The decision to use antibiotics were guided by their previous experience 71% (153/215) and in consultation with other professionals (28%, 61/215) (S3 Table). The respondents with certificate/diploma mentioned that they seek the advice of specialist while using antibiotics compared to those with bachelors/masters (p-value 0.026).

When asked about the antibiotic availability and expiry problem, 46% (102/219) of the respondents have mentioned that they faced shortage of antibiotics in their centers while 71% respondents mentioned that the antibiotic expiry is a problem in their respective centers, and 12% of them had to use expired antibiotics in times of need.

### Factors associated with good knowledge, attitude and practice on antibiotic use and AMR

The final multivariable logistic regression showed that respondents who had read Bhutan national plan on AMR were 2.39 times more likely (AOR: 2.39; 95% CI: 1.19–4.82) to have good knowledge on antibiotics and AMR (Table 4). The female respondents (AOR: 2.16; 95% CI: 1.01–4.61), respondents from the eastern region (AOR: 2.53; 95% CI: 1.18–5.44), west central region (AOR: 3; 95% CI: 1.30–6.92), animal health supervisors (AOR: 9.77; 95% CI: 1.98–48.29) and livestock production supervisors (AOR: 2.77; 95% CI: 1.21–6.35) were significantly associated with favorable attitude towards antibiotics and AMR.

### Discussion

This study was conducted to understand the knowledge, attitude and practice of antibiotics use and AMR among animal health workers in Bhutan. In addition, the study explored the

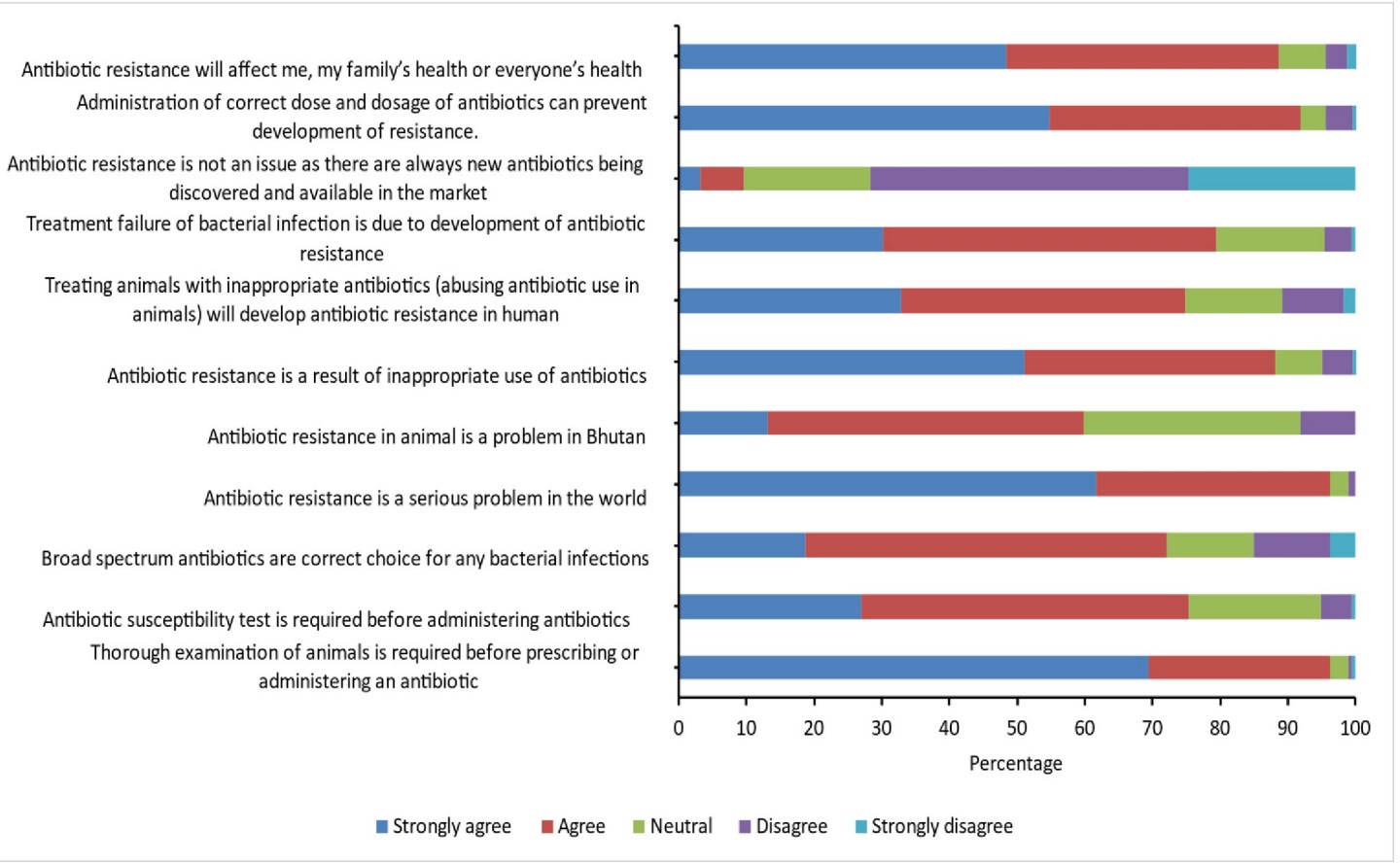

**Fig 3. Attitude on antibiotic usage and AMR among the 219 respondents (animal health workers in Bhutan).**

respondents' awareness on AMR and their prescribing behavior. To the best our knowledge, this is the first study conducted to understand the level of KAP among animal health workers in Bhutan.

Our study demonstrates poor knowledge on antibiotics and AMR among animal health workers in Bhutan with only 38.8% of them scoring above the mean knowledge score. This could be due to low level of awareness among the respondents as can be seen from the awareness data. Of the total respondents only 31 and 34 of them had attended refresher course and had participated in the world AMR awareness week. The low knowledge on antibiotic usage and AMR can lead to unnecessary prescriptions of antibiotics. It was observed that those who read Bhutan national plan on AMR had better knowledge than those who did not, suggesting that the national plan should be distributed to the districts/sub-district animal health workers.

Although, less than 40% of the respondents scored below the mean knowledge score, more than 90% of the respondents gave correct answers for eight of the 17 knowledge questions. Better response rate was recorded when questions were more general such as when asked on the importance of right dose and dosage of antibiotics to be administered, course completion of antibiotics, prophylactic and metaphylactic treatments. However, the response rate became poor when more specific questions were asked. The respondents were not clear on the duration of antibiotics especially for poultry with some mentioning as 3–5 days while some did not know the answer. Although, duration of antibiotic course is a debatable subject at the moment, over dosing and under dosing of antibiotics is one of the factors for development of resistance [19]. It is also worrying to note a quarter of them mentioning that antibiotics can be used in all

**Table 3. Practices on antibiotic prescription among the veterinarians and para-veterinarians.**

| Characteristics | Frequency (%) | Certificate/diploma (%) | Bachelor/Masters (%) | p-value |
|---|---|---|---|---|
| **What type of animal cases do you prescribe antibiotics?** | | | | |
| Any type of case | 24(11) | 18 (11.3) | 6 (10.2) | 0.929 |
| Animals with rhinitis | 35(16) | 27 (16.9) | 8 (13.6) | |
| Animals with diarrhoea | 19(8.7) | 15 (9.4) | 4 (6.8) | |
| animals with inappetence | 4(1.8) | 3 (1.9) | 1 (1.7) | |
| Prolonged illness with underlying bacterial illness* | 137(62.6) | 97 (60.6) | 40 (67.8) | |
| **I make my own judgement to prescribe or administer antibiotics to animals and is not influenced by the animal owner's decision (not based on what owners want)** | | | | |
| Yes* | 180(82.2) | 130 (81.3) | 50 (84.8) | 0.549** |
| No | 39(17.8) | 30 (18.8) | 9 (15.3) | |
| **How many days do you usually prescribe (duration of antibiotic treatment) antibiotics for an animal?** | | | | |
| One day | 2(0.9) | 1 (0.6) | 1 91.7) | 0.003 |
| Three days | 55(25.1) | 49 (30.6) | 6 (10.2) | |
| Five days* | 138(63) | 90 (56.3) | 48 (81.4) | |
| Seven days* | 20(9.1) | 16 (10) | 4 (6.8) | |
| Until the animal recovers | 4(1.8) | 4 (2.5) | 0 | |
| **I think of the type of bacteria involved in infection of an animal before selecting an antibiotic** | | | | |
| Yes* | 187(85.4) | 139 (86.9) | 48 (81.4) | 0.305** |
| No | 32(14.6) | 21 (13.1) | 11 (18.6) | |
| **I give large dose of antibiotics to large animal and small dose to small animal (e.g., Poultry birds) by looking at the size of the animals than the recommended dose** | | | | |
| Yes | 41(18.8) | 30 (18.9) | 11 (18.6) | 0.970** |
| No* | 176(81.2) | 129 (81.1) | 48 (81.4) | |

*Correct practice.

**chi square test.

types of infection. In addition, varying responses were given for answers to the case scenario questions in which most of the respondents chose to prescribe antibiotics even though antibiotics were not required. These findings indicate that the respondents in this study had poor knowledge in these areas and could risk irrational and unintended use of antibiotics in animals that could accelerate resistant development in the country. This suggests a need for more in depth and nationwide training and awareness program to the animal health workers so that sufficient knowledge can be acquired to prevent misuse of antibiotics.

In concurrent to other studies [20, 21], most of the respondents in this study believed that inappropriate use of antibiotics as the drivers of AMR. However, it is worrying to note that most respondents (72%) agreeing on the use of broad-spectrum antibiotics for any bacterial infection. This could be due to lack of adequate knowledge on antibiotics and data on antibiotic sensitivity test and testing facilities in the country, as indicated by low agreement to need for sensitivity test (75%) compared to other studies [20, 21]. Inappropriate use of antibiotics has been identified as one of the leading causes of AMR [1]. This calls for the proper education on the causes of AMR and improvement on the basic antibiotic sensitivity testing facility in the country for prevention of AMR. Although, most of the respondents (96%) in this study believed antibiotic resistance as a serious global problem, the fact that only 59% believed it as a national problem could be linked to lack of awareness as only about 37% of them read the national plan on AMR. This is in contrast to findings from Nigeria where more than 90% of the respondents believed AMR as a national problem [13].

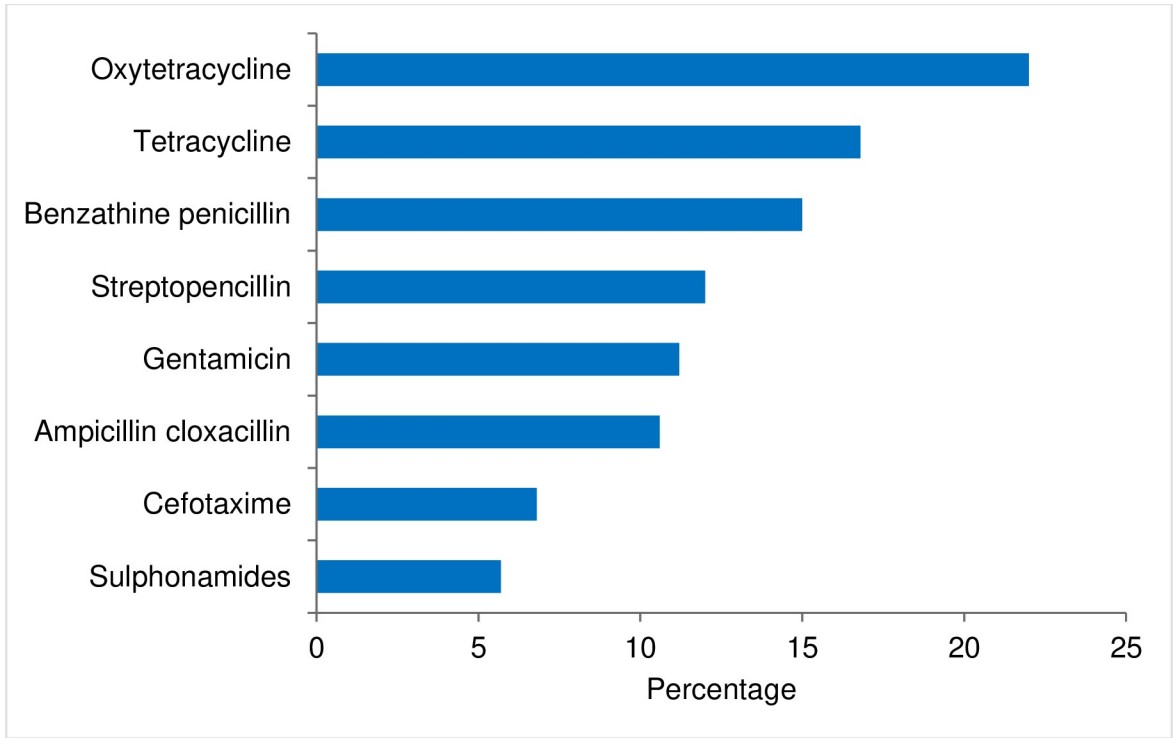

**Fig 4. Most used antibiotics in Bhutan based on the information provided by the 219 animal health workers (respondents).**

**Table 4. Factors associated with good knowledge, attitude and practice on antibiotic use and AMR among the veterinarian and para-veterinarians in Bhutan.**

| Categories | Variable/ Categories | AOR (95% CI) | p-value |
|---|---|---|---|
| | *Read on national plan on AMR* | | |
| | No | Reference | |
| Knowledge | Yes | 2.39 (1.19–4.82) | 0.015 |
| | *Gender of the respondents* | | |
| | Male | Reference | |
| | Female | 2.16 (1.01–4.61) | 0.047 |
| | *Region where the respondents work* | | |
| | West | Reference | |
| Attitude | East | 2.53 (1.18–5.44) | 0.017 |
| | East Central | 2.44 (0.85–6.96) | 0.096 |
| | West central | 3 (1.30–6.92) | 0.01 |
| | *Current position of the respondents* | | |
| | Extension supervisor | Reference | |
| | Animal Health supervisor | 9.77 (1.98–48.29) | 0.005 |
| | Gewog livestock extension officer | 1.72 (0.78–3.78) | 0.181 |
| | Livestock production officer | 6.44 (0.57–72.81) | 0.132 |
| | Livestock production supervisor | 2.77 (1.21–6.35) | 0.016 |
| | Veterinarian | 2.33 (0.78–6.97) | 0.13 |

Almost 72% of the respondents in this study believed that there are no new antibiotics available to combat antibiotic resistance. This perception towards antibiotic use can prevent unnecessary prescription of antibiotics to prevent resistance. This is much higher compared to studies in Nigeria (29%) [13] and Iran (35%) [21]. The difference could be due to difference in the study population. Whereas in Iran, the respondents being students with less exposure to clinical practice would have affected this belief.

In contrast to the knowledge, the overall practices on antibiotic usage were better which could be due to experience and availability of treatment guidelines as almost all respondents used veterinary drug formulary distributed by the government for use in the field. However, although, all the respondents (100%) have answered (have knowledge) that antibiotics should be administered with correct dose and dosage for any animal species but in practice, 18.8% (n = 41) of the respondents have mentioned that they give large dose of antibiotics to large animal and small dose to small animals. Decision to prescribe or not was influence by their previous experience (71%) and consultation with other animal health professionals (28%). The majority of our respondents' decision to prescribe was not influenced by the clients which is a good indication as our animal health workers will not be under pressure to prescribe in cases where antibiotics are not required. This finding was similar to studies from the US [22] and Australia [23]. This would prevent unnecessary prescription of antibiotic for a disease under the pressure of the clients. It is also reassuring to note that most of the respondents thought of the type of bacteria involved in an infection before selecting an antibiotic. This would help in choosing the right antibiotics against an organism causing that infection. However, it is worrying to see that some of our respondents prescribed antibiotics to any type of cases, rhinitis, diarrhea and inappetence where it was not required. This suggests there are misuse of antibiotics in practice that could accelerate the development of AMR in the country.

Although, the overall level of practice did not differ between bachelors/master and certificate/diploma, there was significant difference in duration of antibiotics prescribed. Those holding certificates/diplomas prescribed antibiotics for shorter duration (3 days) days compared to bachelors/master that prescribed for the recommended 5 days. This difference could be difference in place of practice as most bachelors/master's holders are in district/city veterinary hospitals that deal with pet animals. Pet animals are easy to follow up for antibiotic therapy compared to farm animals that are isolated and difficult to reach. Those with certificate/diploma holders mostly work with farm animals, and moreover the livestock farming in Bhutan are small holders, scattered and difficult to reach for follow up treatment. Duration of antibiotic course is one of the factors contributing to AMR. However, there is lack of evidence to suggest for the right duration of antibiotic course without the local/regional data [19].

Similar to many other studies [23–25], oxytetracycline, tetracycline and penicillin were the most commonly prescribed antibiotics in animals in Bhutan. This could be due to their availability, familiarity among the animal health workers, and the disease prevalence in the country. Like in many other studies [26, 27], most respondents' (>90%) antibiotic prescription was guided by standard guidelines on antimicrobial usage. The guidelines outline the antibiotic availability in Bhutan as well as the indications of it based on the national record on the sensitivity pattern and international reports. One method to combat increasing drug resistance has been the implementation of guidelines and proper regulation of its usage. It has been shown that introduction of antibiotic guideline led to significant reduction of antibiotic prescription and also led to use of lower generation drugs [27]. Therefore, one way of rationalization of antibiotic use could be introduction of antimicrobial guidelines.

The choice of antibiotics in our study was determined by the availability of it in the center in contrast to other studies where in the choice of antibiotics was dependent on clients 'affordability [13]. In Bhutan, all types of veterinary services including antibiotics and all other

medicines are provided free of cost and are regulated by the government agencies. This type of practice can limit the misuse of antibiotics. It can also lead to inappropriate use of antibiotics due to mismatch between disease and the available antibiotics at the veterinary centers. Depending on supply from central agencies, it would also lead to shortage of antibiotics in the veterinary centers leading to use of expired drugs as indicated in this study where in 12% of the respondents used expired medicines. This calls for a need at the department level to provide a regulated supply of medicine based on the disease pattern in the country. Generally, in the field centers, all the disease conditions are entered and submitted in the online database, which is accessible to the central agencies. Therefore, the procurement agencies and the field staff need to work in hand to hand to rectify these issues. Therefore, the Government should aim to stock all the required antibiotics with proper guidelines for its usage.

## Conclusions

Our study identified that most animal health workers in Bhutan had poor knowledge on antibiotics use and AMR. The choice of antibiotic was influenced by their availability in the centre. This could lead to inappropriate use of antibiotics. Hence interventions are needed to make the required antibiotics available based on the sensitivity data and disease epidemiology. Most importantly, regular awareness education on antibiotics and AMR in the form of refresher course/training and workshop must be provided to the animal health workers in the country.

## Supporting information

**S1 Table. Bivariate analysis of knowledge on antibiotics and AMR among the veterinarian and para-veterinarians categorized by the level of qualification.**
(DOCX)

**S2 Table. Attitude towards antibiotics and AMR among the veterinarian and para-veterinarians in Bhutan.**
(DOCX)

**S3 Table. Prescribing pattern of antibiotics among the veterinarian and para-veterinarians in Bhutan.**
(DOCX)

**S1 Text. Questionnaire for KAP survey on antibiotics and AMR.**
(PDF)

**S1 File.**
(DTA)

## Acknowledgments

We thank the veterinarians and para-veterinarians in the country for their participation in the survey and the dzongkhag livestock officers for facilitating the data collection.

## Author Contributions

**Conceptualization:** Karma Wangmo, Thinley Dorji, Tenzin Tenzin.

**Data curation:** Karma Wangmo, Thinley Dorji.

**Formal analysis:** Karma Wangmo, Thinley Dorji, Tenzin Tenzin.

**Investigation:** Karma Wangmo, Thinley Dorji, Tshering Dorji, Jambay Dorji.

**Methodology:** Karma Wangmo, Thinley Dorji, Tshering Dorji, Tenzin Tenzin.

**Project administration:** Karma Wangmo.

**Resources:** Karma Wangmo.

**Software:** Karma Wangmo, Thinley Dorji.

**Supervision:** Tenzin Tenzin.

**Validation:** Karma Wangmo, Thinley Dorji, Jambay Dorji, Tenzin Tenzin.

**Visualization:** Karma Wangmo, Narayan Pokhrel.

**Writing – original draft:** Karma Wangmo.

**Writing – review & editing:** Karma Wangmo, Thinley Dorji, Narayan Pokhrel, Tshering Dorji, Jambay Dorji, Tenzin Tenzin.

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
