## [Decision Letter · Decision Letter 0]

12 Feb 2021

PONE-D-20-40585

Knowledge, attitude, and practice on antibiotic use and antibiotic resistance among the veterinarians and para-veterinarians in Bhutan

PLOS ONE

Dear Dr. Wangmo,

Thank you for submitting your manuscript to PLOS ONE. After careful consideration, we feel that it has merit but does not fully meet PLOS ONE’s publication criteria as it currently stands. Therefore, we invite you to submit a revised version of the manuscript that addresses the points raised during the review process.

The reviewers have raised a number of questions that need clarification and better explanation. Please address all reviewer comments point by point. 

We look forward to receiving your revised manuscript.

Kind regards,

Iddya Karunasagar

Academic Editor

PLOS ONE

Journal Requirements:

2. Thank you for including your ethics statement:  "The study was approved by the Research and Extension Division of the Department of Livestock, Ministry of Agriculture and Forest, Bhutan on 21 April 2020.".   

PRTC request:(either IRB not named or confirm approval)

Closing sentence:

3. Please ensure that you refer to Figure 1 and 2 in your text as, if accepted, production will need this reference to link the reader to the figure.

Additional Editor Comments:

Reviewers have raised number of points that need clarification and better explanation. These relate to various sections including methodology used. Please respond to reviewer comments point by point.

Reviewers' comments:

Reviewer's Responses to Questions

**Comments to the Author**

1. Is the manuscript technically sound, and do the data support the conclusions?

Reviewer #1: Yes

Reviewer #2: Yes

2. Has the statistical analysis been performed appropriately and rigorously? 

Reviewer #1: I Don't Know

Reviewer #2: Yes

3. Have the authors made all data underlying the findings in their manuscript fully available?

Reviewer #1: Yes

Reviewer #2: Yes

4. Is the manuscript presented in an intelligible fashion and written in standard English?

Reviewer #1: Yes

Reviewer #2: Yes

5. Review Comments to the Author

Reviewer #1: This is a very important paper on a very interesting topic.I applaud the authors for taking up this challenging work. However, there are several areas in the manuscript that need improvement, clarification, and editing.Line 28: AMR… veterinarians/para-veterinarians in Bhutan. It must come as Veterinarians and para-veterinarians Line 49-50: Without proper knowledge how come they can do good practice? According to me, it is merely a speculative statement. The findings partially fail to support the statement. Line 113-114:  antibiotic prescribing trends……para-veterinarians – Does Bhutan country allows para-veterinarians to prescribe an antibiotic? Have they qualified adequately in par with Veterinarians to prescribe antibiotics?Line 143: online using google ……. HGA). The link is closed. I recommend the authors to attach a copy of the unfilled google form/questionnaires an attachment/annexure. Line 172-174: Mention the reference number of ethical approval letter. Line 196-197: All the ….. animal species. Have you confirmed/cross verified this finding? If so, how?Table 2, Pg no: 13: Any bacteria ……. Antibiotic. The question is non-practical. What will happen if the bacteria exposed to the antibiotic at the right dose and right duration will become resistant?Table 2, Pg no: 13: A resistant bacterium……… also to humans.May I know the reason why the authors have forgotten the environmental aspect as well as the agricultural aspect in the transmission part? In general, I recommend good copy editing to fix grammatical errors and punctuation.

Reviewer #2: Overall, this is a nicely done study that highlights the importance of information, education and communication related to antibiotics amongst the animal health care workers. Some minor comments:

Line 114: “an” evidence based.

Line 127: you mean “para-veterinarians.”

Line 132: data “was” collected.

Line 135-136: either write “17” or “Seventeen”; “4” or “four”. Please be consistent. Somewhere it is a spelling and somewhere it is a number.

Line 142: Instead of “latter” write “later.”

Line 146: mention the “eligibility criteria” of the participants somewhere.

Line 186: degree in “veterinary science” you mean ?

Line 371-372: Please write 46% instead of “forty-six” and 71 instead of seventy-one.

6. PLOS authors have the option to publish the peer review history of their article (what does this mean?). If published, this will include your full peer review and any attached files.

Reviewer #1: No

Reviewer #2: No

---

## [Author Response · Author response to Decision Letter 0]

12 Mar 2021

Response to the Editor and reviewers 

Editorial comments

2. Thank you for including your ethics statement: "The study was approved by the Research and Extension Division of the Department of Livestock, Ministry of Agriculture and Forest, Bhutan on 21 April 2020.". 

PRTC request:(either IRB not named or confirm approval)

Closing sentence:

 Response: We have added the ethical approval number in the text. (Line No: 179-180). In addition, we have included the information regarding the informed consent of the respondents (line no:180-183)

3. Please ensure that you refer to Figure 1 and 2 in your text as, if accepted, production will need this reference to link the reader to the figure.

Response: We have referred the Figure 1 and 2 in the text. (Line No:240 and 249)

Reviewer #1: 

This is a very important paper on a very interesting topic. I applaud the authors for taking up this challenging work. However, there are several areas in the manuscript that need improvement, clarification, and editing.

Line 28: AMR… veterinarians/para-veterinarians in Bhutan. It must come as Veterinarians and para-veterinarians 

Response: Thank you and we have corrected in the revised manuscript. (Line number: 27)

Line 49-50: Without proper knowledge how come they can do good practice? According to me, it is merely a speculative statement. The findings partially fail to support the statement. 

Response: Thank you for pointing this out. The sentence has been corrected as per the findings (Line: 48-49)

Line 113-114: antibiotic prescribing trends……para-veterinarians – Does Bhutan country allows para-veterinarians to prescribe an antibiotic? Have they qualified adequately in par with Veterinarians to prescribe antibiotics?

Response: 

Yes, the para-veterinarians in Bhutan are allowed to prescribe antibiotics. The para-veterinarians in Bhutan have a formal two-year diploma in animal husbandry (after completion of grade 10 and 12) where they are taught veterinary medicines and pharmacology in their curriculum. They are also provided refresher course in veterinary medicine and veterinary drugs time to time. They also seek technical support from the veterinarians and would refer the Drug Formulary developed by the Department wherever necessary. In addition, the Drug Regulatory Authority of Bhutan conduct constant monitoring of the antibiotic prescription in the field. 

We have included this information in the revised manuscript. (Line: 126-133)

Line 143: online using google ……. HGA). The link is closed. I recommend the authors to attach a copy of the unfilled google form/questionnaires an attachment/annexure.

Response: The questionnaire form has been attached as supplementary file (Line No: 646) 

 Line 172-174: Mention the reference number of ethical approval letter. 

Response: We have added the ethical approval number in the revised manuscript. Line number: 179-180)

Line 196-197: All the ….. animal species. Have you confirmed/cross verified this finding? If so, how? 

Response: Yes, we have cross verified this finding with another question on their practice on antibiotic with a question “I give large dose of antibiotics to large animal and small dose to small animal (e.g., Poultry birds) by looking at the size of the animals than the recommended dose” (Table 3). Although, all the respondents (100%) have answered (have knowledge) that antibiotics should be administered with correct dose and dosage for any animal species but in practice, 18.8% (n=41) of the respondents have mentioned that they give large dose of antibiotics to large animal and small dose to small animals (see Table 3). We have discussed these points in the discussion. (Line No 464-467)

Table 2, Pg no: 13: Any bacteria ……. Antibiotic. The question is non-practical. What will happen if the bacteria exposed to the antibiotic at the right dose and right duration will become resistant? 

Response: Yes, you are correct. However, we have kept this question/answer (finding) as it is. 

Table 2, Pg no: 13: A resistant bacterium……… also to humans. May I know the reason why the authors have forgotten the environmental aspect as well as the agricultural aspect in the transmission part?In general, I recommend good copy editing to fix grammatical errors and punctuation.

Response: Thank you for your comments and suggestions. Unfortunately, we have forgotten to include the environmental and the agricultural aspect in the transmission part in this study. We will look into this aspect in future studies. 

Reviewer #2:

Overall, this is a nicely done study that highlights the importance of information, education and communication related to antibiotics amongst the animal health care workers. Some minor comments:

Response: Thank you for providing critical comments/suggestions to improve our manuscript. We have addressed all your suggestions in the revised manuscripts. 

Line 114: “an” evidence based.

Response: We have corrected this in the revised manuscript. (Line number: 111)

Line 127: you mean “para-veterinarians.”

Response: We have corrected this in the revised manuscript (Line number:124)

Line 132: data “was” collected.

Response: We have corrected this in the revised manuscript (Line number: 136)

Line 135-136: either write “17” or “Seventeen”; “4” or “four”. Please be consistent. Somewhere it is a spelling and somewhere it is a number.

Response: Thank you for your suggestion. We have changed it as suggested in the revised manuscript (Line number: 139)

Line 142: Instead of “latter” write “later.”

Response: Thank you for correction and we have revised it (Line number 146)

Line 146: mention the “eligibility criteria” of the participants somewhere.

Response: Thank you for the suggestion. We have added the eligibility criteria in the revised manuscript as the government veterinarians and para-vets working in the field, and were handling/prescribing antibiotics for the treatment of animal cases. (Line number: 147-149)

Line 186: degree in “veterinary science” you mean?

Response: The qualification for degree is veterinary science for veterinarians (5 years degree course) and diploma (2-year course) in animal science for para-veterinarians. Some of the para-veterinarians have upgraded their qualification to degree (3 years degree) in animal sciences. We have added this in the revised manuscript (line number:195-196)

Line 371-372: Please write 46% instead of “forty-six” and 71 instead of seventy-one.

Response: We have corrected this in the revised manuscript. (line number: 382-383)

---

## [Decision Letter · Decision Letter 1]

26 Apr 2021

Knowledge, attitude, and practice on antibiotic use and antibiotic resistance among the veterinarians and para-veterinarians in Bhutan

PONE-D-20-40585R1

Dear Dr. Wangmo,

We’re pleased to inform you that your manuscript has been judged scientifically suitable for publication and will be formally accepted for publication once it meets all outstanding technical requirements.

Kind regards,

Iddya Karunasagar

Academic Editor

PLOS ONE

Additional Editor Comments (optional):

All reviewer comments have been addressed satisfactorily.

Reviewers' comments:

Reviewer's Responses to Questions

**Comments to the Author**

1. If the authors have adequately addressed your comments raised in a previous round of review and you feel that this manuscript is now acceptable for publication, you may indicate that here to bypass the “Comments to the Author” section, enter your conflict of interest statement in the “Confidential to Editor” section, and submit your "Accept" recommendation.

Reviewer #1: All comments have been addressed

2. Is the manuscript technically sound, and do the data support the conclusions?

Reviewer #1: Yes

3. Has the statistical analysis been performed appropriately and rigorously? 

Reviewer #1: I Don't Know

4. Have the authors made all data underlying the findings in their manuscript fully available?

Reviewer #1: Yes

5. Is the manuscript presented in an intelligible fashion and written in standard English?

Reviewer #1: Yes

6. Review Comments to the Author

Reviewer #1: (No Response)

7. PLOS authors have the option to publish the peer review history of their article (what does this mean?). If published, this will include your full peer review and any attached files.

Reviewer #1: **Yes: **Thulasiraman Parkunan

---

## [Editor Report · Acceptance letter]

28 Apr 2021

PONE-D-20-40585R1 

Knowledge, attitude, and practice on antibiotic use and antibiotic resistance among the veterinarians and para-veterinarians in Bhutan 

Dear Dr. Wangmo:

I'm pleased to inform you that your manuscript has been deemed suitable for publication in PLOS ONE. Congratulations! Your manuscript is now with our production department. 

Kind regards, 

on behalf of

Dr. Iddya Karunasagar 

Academic Editor

PLOS ONE